# On Gaussian Mixture Models

## ABSTRACT

We investigate the sample complexity of Gaussian mixture models (GMMs). Our results provide the optimal upper bound, in the context of uniform spherical Gaussian mixtures. Furthermore, we highlight the relationship between the sample complexity of GMMs and the distribution of spacings among their means.

Gaussian Mixture Models have been extensively studied in the field of machine learning. Let $\mathcal{N}$ represent the Probability Density Function (PDF) of a multidimensional Gaussian distribution characterized by a mean $\mu$ and a covariance matrix $\Sigma$. A Gaussian Mixture Model is defined as follows:

$$\Gamma = \sum_{i=1}^{k} \omega_i \mathcal{N}(\mu_i, \Sigma_i), \tag{0.1}$$

where $\sum \omega_i = 1$.

Assume that we are given a sequence of i.i.d. samples, $x_1, \cdots, x_n$, generated from $\Gamma$. An important area of research on GMM's is determining the optimal number of samples required to learn the underlying mixture.

Learning the mixture can be divided into two distinct approaches. The first approach aims to yield a distribution that closely approximates $\Gamma$, in terms of total variation (2). However, this approach doesn't inherently offer insights into the specific parameters themselves. The second approach involves the estimation of each individual parameter of $\Gamma$, namely $\omega_i$, $\mu_i$, and $\Sigma_i$, with a "good precision". To elaborate, let's clarify the term "good precision".

**Definition 1** *For a given $\Gamma$ as in equation 0.1, an algorithm is said to learn the parameters of $\Gamma$ with $\epsilon$-precision if, for every $\mu_i, \Sigma_i$, and $\omega_i$ in equation 0.1, it produces $\hat{\mu}_i, \hat{\Sigma}_i$, and $\hat{\omega}_i$ such that $|\mu_i - \hat{\mu}_i| < \epsilon$, $|\Sigma_i - \hat{\Sigma}_i| < \epsilon$, and $|\omega_i - \hat{\omega}_i| < \epsilon$.*

Parameter estimation of a GMM dates back to the 18th century and was initially introduced by Pearson(12). Subsequently it has been studied in various aspects (9; 4; 3; 6; 7; 8). If the parameters of the mixture all well separated, it can be shown that around $\epsilon^{-2}$ samples are sufficient to estimate parameters with $\epsilon$-precision (9; 3). Despite the rich body of work on this problem there are not many results that link the parameter distribution to sample complexity. This constitutes a central theme of the present paper.

Our work is inspired by the paper of Hardt and Price (9). They explored the case of $k = 2$ and demonstrated that $\epsilon^{-12}$ samples are both necessary and sufficient for determining the parameters of a 2-mixture with $\epsilon$-precision. If the means are well-separated, merely $\epsilon^{-2}$ samples are enough.

For $k = 2$, there are 5 parameters to learn. Exponent 12 suggests that we need a factor of $\epsilon^{-2}$ samples to distinguish between each of these parameters. When this is done, with an extra $\epsilon^{-2}$ samples we can reach $\epsilon$-accuracy.

For arbitrary $k$, there are $3k - 1$ parameters to learn. Hardt and Price (9), considered a $k$-mixture with $3k - 2$ parameters [1], and showed that at least $\epsilon^{-6k+2}$ samples are necessary for this case. This leads us to anticipate the optimal lower bound, for the general case, to be $\epsilon^{-6k}$, which raises the following question: are $\epsilon^{-6k}$ i.i.d. samples adequate for determining the parameters of a GMM with $\epsilon$-precision?

**Our Contribution.** We answer the above question affirmatively, in the uniform spherical case, where $\omega_i = 1/k$ and $\sigma_i = 1$. Further, we establish a connection between pair correlation of the means of the mixture ($\mu_i$'s) and the sample complexity of learning its parameters. In essence, we prove that the distribution of spacing between the $\mu_i$'s greatly influences the sample complexity. This connection is a novel part of this paper.

Note that in the uniform spherical case with mean zero, we have $k - 1$ parameters to recover.Therefore, we expect $\epsilon^{-2k}$ samples to suffice.

**Corollary 0.1** *Let $\Gamma$ be a uniform spherical mixture of $k$ Gaussians distribution with mean equal to zero and variance equal to $\sigma^2$. Assume that $\epsilon < \min_{i \neq j}(|\mu_i - \mu_j|)$, and let $c(\sigma, k) = \left| k^2(1 + k\sigma^2)^{k/2}(2\sigma)^k e^{0.5(k/\sigma)^2} \right|^2$ . Then*

$$10^4 c(\sigma, k)\epsilon^{-2k}$$

*samples are sufficient to learn parameters of $\Gamma$ with $\epsilon/100$-accuracy.*

Corollary 0.1 shows that in the worst case scenario, we requires $\epsilon^{-2k}$ samples to learn parameters with $\epsilon$-accuracy. However, this is not always the case. There are many instances where we can learn parameters of a GMM with fewer samples. Next, we demonstrate how the sample complexity is contingent upon the distribution of spacing between the mixture's means.

## 0.1 PAIR CORRELATION OF MEANS AND SAMPLE COMPLEXITY.

Hardt and Price (9) have shown that if $|\mu_1 - \mu_2| = \Omega(\sigma)$, roughly $\epsilon^{-2}$ samples are sufficient for determining the parameters of a 2-mixture with $\epsilon$-precision. In general, without this separation, $\epsilon^{-12}$ samples would be required. This implies that pair correlations of means can impact the sample complexity. We expand on this phenomenon for general $k$.

To explain the pair correlation's role, assume that we have gaps of length $\epsilon$ between consecutive means in our mixture. However, these gaps are isolated, meaning that if $\mu_{n+1} - \mu_n = \epsilon$, then the adjacent gaps are significantly larger: $\mu_{n+2} - \mu_{n+1} \gg 1$, and $\mu_n - \mu_{n-1} \gg 1$. Further assume that we are aiming to learn $\mu_i$'s with $\epsilon/100$-accuracy. In this situation our result shows having roughly $c(k, \sigma)\epsilon^{-4}$ samples will suffice.

Now let us alter the situation by allowing two consecutive gaps of length $\epsilon$ in our mixture .i.e. $\mu_{n+1} - \mu_n = \epsilon$, and $\mu_n - \mu_{n-1} = \epsilon$. As before, we assume that other surrounding gaps remain significantly larger: $\mu_{n+2} - \mu_{n+1} \gg 1$, and $\mu_{n-1} - \mu_{n-2} \gg 1$. Under this modified scenario, our required sample complexity swells to $c(k, \sigma)\epsilon^{-6}$.

Let us define a function that captures the essence of the pair correlation for our purpose. Let

$$\mathcal{P}(\mu_m) = \prod_{\substack{n=1 \\ n \neq m}}^{k} |\mu_m - \mu_n|. \tag{0.2}$$

When an $\epsilon$-vicinity of $\mu_m$ contains $l$ means, then $\mathcal{P}(\mu_m) \approx \epsilon^l$. Our next corollary shows the maximum number of means that are located in an interval of length about $\epsilon$ determines the sample complexity.

---

[1]They assume that at least two of the parameters cannot be very close to each other, which saves a factor of $\epsilon^{-2}$ samples. See (9) the paragraph before Theorem 2.10.

**Corollary 0.2** *Let $\Gamma$ be a uniform spherical mixture of $k$ Gaussians distribution with mean equal to zero and variance equal to $\sigma^2$. Then, given*

$$\log\left(\frac{1}{\delta}\right)\frac{c(\sigma, k)}{\left|\min_m \mathcal{P}(\mu_m)\right|^2}\,\epsilon^{-2}, \tag{0.3}$$

*samples from $\Gamma$, where $c(\sigma, k) = \left|k^2(1+k\sigma^2)^{k/2}(2\sigma)^k e^{0.5(k/\sigma)^2}\right|^2$, with probability $1-\delta$, we can approximate $\Gamma$'s parameters with $\epsilon$-precision.*

Next we state our main theorem.

**Theorem 0.3** *Let $\Gamma = \frac{1}{k}\sum_{m=1}^{k}\mathcal{N}(\mu_m, 1)$, where its mean is zero and variance is $\sigma^2$. Assume that we are given $\log\left(\frac{1}{\delta}\right)\epsilon^{-2}$ samples from $\Gamma$ with*

$$\epsilon < \frac{\mathcal{P}(\mu_m)}{k^2(1+|\mu_m|^2)^{(k-1)/2}(2\sigma)^k e^{0.5(k/\sigma)^2}}.$$

*Then our algorithm returns $\hat{\mu}_m$ such that*

$$|\hat{\mu}_m - \mu_m| < \frac{k^2(1+|\mu_m|^2)^{k/2}(2\sigma)^k e^{0.5(k/\sigma)^2}}{\mathcal{P}(\mu_m)}\epsilon. \tag{0.4}$$

## 0.2 RELATED WORK.

We have already mentioned the work of Hardt and Price (9). Parallels to our approach, they too employ the method of moments. Their scope, however, is constrained to mixtures of two Gaussians. Similarly to our work, a main aspect of their work revolves around the correlation between the separation of parameters and sample complexity.

A substantial separation between parameters intuitively simplifies the task of clustering samples. In this genre, for the Spherical GMM, Liu and Li (3) proposed an algorithm that operates within a $poly(d, k)$ time frame. Their result is contingent on the condition:

$$|\mu_i - \mu_j| > (\log k)^{1/2+\varepsilon}$$

where $d$ signifies the dimension. It's noteworthy that when the separation between parameters exceeds 1, our algorithm attains $\epsilon$-accuracy with approximately $\epsilon^{-2}$ samples.

For scenarios where $d < \log k$, Qiao et al. (4) further refined the results of Liu and Li. Their sample complexity though, with respect to $\epsilon$-precision, is quasi-polynomial i.e., $\epsilon^{-c\log(1/\epsilon)}$.

What sets our method apart is the absence of assumptions regarding parameter separation, and the optimal bound on sample complexity subject to accuracy. Additionally, we clearly describe the circumstances under which more than $\epsilon^{-2}$ samples—a theoretical minimum—are required.

## 0.3 PROOF OVERVIEW

Let $P$ be a polynomial of degree $k$ with roots corresponding to the parameters of our mixture; we'll refer to $P$ as our "parameter polynomial". We aim to approximate coefficients of $P$ with a good precision and relate this approximation to the number of samples we use.

We employ the method of moments combined with Newton's identity to derive a polynomial whose coefficients are close to the coefficients of $P$. In practice, since we are using samples to calculate the moments of our mixture, we need to consider the empirical error. Therefore, we need to have a sense of how much of this error would spill into our approximation of the parameter polynomial. Following this process we end up with a polynomial whose coefficients are within a certain distance

to $P$. This distance is proportional to the number of samples we use to calculate empirical moments.

Lastly, we must establish an argument that connects the similarity between the coefficients of two polynomials to the difference between their roots. Here we use a theorem from real analysis.

## 1 THE METHOD OF MOMENTS

One way of approaching the problem of parameters estimation of GMM's, is through the method of moments, see (6; 7; 9). For a one dimensional Gaussian we have the $r$-th moment equals

$$\mathcal{M}_r(\mu, \sigma) := \frac{1}{\sigma\sqrt{2\pi}} \int x^r e^{-\frac{1}{2}\left(\frac{x-\mu}{\sigma}\right)^2}, \tag{1.1}$$

which is a polynomial in terms of $\mu, \sigma$ and easily calculable.

For a $k$-mixture, to simplify the problem, assume that $\omega_i = 1/k$, and $\Sigma_i = \sigma_i \in \mathbb{R}$. Therefore, moments of the $k$-mixture are

$$M_m = \frac{1}{k}\sum_{i=1}^{k} \mathcal{M}_d(\mu_i, \sigma_i) = \frac{1}{k}\sum_{i=1}^{k}\sum_{j=0}^{\lfloor m/2 \rfloor} c_{m,j}\mu_i^{m-2j}\sigma_i^{2j}. \tag{1.2}$$

Here we have $2k$ variables $\mu_1\cdots, \mu_k$ and $\sigma_1\cdots, \sigma_k$, and as many equations as we desire. There are various ways to solve a system of equations, however the first barrier here is that we can only approximate $M_i$, using samples.

Given $n$ i.i.d. samples $x_1\cdots x_n$ from our mixture, the $m$-th empirical moment is

$$\hat{M}_m = \tfrac{1}{n}\sum_{j=1}^{n} x_j^m. \tag{1.3}$$

Note that $\hat{M}_m$ is random variable with mean $M_i$ and variance $\sigma^{2m}/n$, where $\sigma^2$ is the variance of the $k$-mixture. By Chebyshev's inequality we have

$$\mathbb{P}\big[|\hat{M}_m - M_m| > r\frac{\sigma^m}{\sqrt{n}}\big] < \frac{1}{r^2}. \tag{1.4}$$

Let us take $n \geq \epsilon^{-2}$, then the portion of samples for which we have $|\hat{M}_m - M_m| > r\epsilon\sigma^m$, is bounded by $r^{-2}$. This would suffice for our purposes, take for example $r = 10$, then we know that for 99% of samples we have

$$|\hat{M}_m - M_m| < 10\epsilon\sigma^m.$$

This can be improved by taking samples in groups, calculate $\hat{M}_m$ for each group and look at the median of these estimates. In (5)[Lemma 3.2] Hardt and Price proved that given $n \gg \log(1/\delta)\epsilon^{-2}$, samples from the mixture, with probability $1 - \delta$ we have:

$$|\hat{M}_i - M_i| < \epsilon\sigma^i. \tag{1.5}$$

Going forward we will use equation 1.5.

## 2 THE UNIFORM SPHERICAL CASE

In this section we prove some necessary lemmas that will help us recover parameters of the mixture. We assume that all variances are equal to $1$. Our final goal is to estimate coefficients of the following polynomial:

$$P(x) := \prod_{i=1}^{k}(x - \mu_i),$$

using information we obtain from equation 1.2. Define

$$P_m(\mu_1, \cdots, \mu_k) = \mu_1^m + \cdots + \mu_k^m.$$

We can write each moments in equation 1.2, in terms of $P_m$ and vice versa.

Let us first, precisely calculate coefficients $c_{m,j}$ in equation 1.2, then we move to approximate $P_m$ using imperial moments we obtain in equation 1.3.

**Lemma 2.1** *For $m \geq 1$ we have that*

$$\mathcal{M}_m(\mu, 1) = \mu^m + \sum_{i=1}^{m/2} \binom{m}{2i} \mu^{m-2i}(2i-1)!!, \tag{2.1}$$

*where $(2i-1)!!$ is the product of odd numbers less than $2i-1$.*

**Remark 1** *Using the lemma we find the expansion of $M_m$ in terms of $P_i$ :*

$$kM_m = P_m + \sum_{i=1}^{m/2} \binom{m}{2i} (2i-1)!!P_{m-2i}. \tag{2.2}$$

*In the next lemma we show that*

$$P_m = kM_m + k \sum_{i=1}^{m/2} (-1)^i \binom{m}{2i} (2i-1)!!M_{m-2i}. \tag{2.3}$$

**Proof. 1 (Proof of Lemma 2.1)** *We have*

$$\mathcal{M}_m(\mu, 1) = \frac{1}{\sqrt{2\pi}} \int x^r e^{-\frac{1}{2}(x-\mu)^2}$$
$$= \frac{1}{\sqrt{2\pi}} \int (x+\mu)^m e^{-\frac{1}{2}x^2} = \sum_i \binom{m}{i} \mu^{m-i} E(x^i),$$

*where $x \sim \mathcal{N}(0,1)$. We use $E(zf(z)) = E(f'(z))$, therefore we have*

$$E(x^m) = E(xx^{m-1}) = (m-1)E(x^{m-2}).$$

*Also note that $E(x^m)$ is zero if $m$ is odd. This will give the lemma.*

We now define a new object that is a empirical approximation of $P_m$.

**Definition 2** *Let $\hat{M}_m$ be as equation 1.3, following equation 2.3, we define*

$$\hat{P}_m = k\hat{M}_m + k \sum_{i=1}^{m/2} (-1)^i \binom{m}{2i} (2i-1)!!\hat{M}_{m-2i}. \tag{2.4}$$

**Lemma 2.2** *We have that $P_m$ satisfy:*

$$P_m = kM_m + k \sum_{i=1}^{m/2} (-1)^i \binom{m}{2i} (2i-1)!!M_{m-2i}. \tag{2.5}$$

*Moreover, Let $\varepsilon_m = \hat{P}_m - P_m$ and $\Delta_m = \hat{M}_m - M_m$. We have that*

$$\varepsilon_m = k\Delta_m + k \sum_{i=1}^{m/2} (-1)^i \binom{m}{2i} (2i-1)!!\Delta_{m-2i}. \tag{2.6}$$

**Proof. 2** *We proceed with using induction, we have that $kM_1 = P_1(\mu_1, \cdots, \mu_k)$ and $kM_2 = 1 + P_2(\mu_1, \cdots, \mu_k)$. This gives us equation 2.5 and equation 2.6 for the base*

*cases. We assume equation 2.6 for $m - 2, m - 4, \cdots$ and we prove it for $m$.*

*Using equation 2.2, we have the following recursive identity relating $\varepsilon_m$ to $\Delta_m$ and $\varepsilon_{m-2}, \varepsilon_{m-4}, \cdots$.*

$$\varepsilon_m = k\Delta_m - \sum_{i=1}^{m/2} \binom{m}{2i}(2i-1)!!\varepsilon_{m-2i}. \tag{2.7}$$

*We use our induction hypothesis in equation 2.7 and we get everything in terms of $\Delta_m, \Delta_{m-2}, \cdots$. Only thing remaining is to calculate the coefficients of $\Delta_{m-2i}$ for $0 < i \leq m/2$.*

*The term $\Delta_{m-2i}$ appears in equation 2.7 expansion of $\varepsilon_m, \varepsilon_{m-2}, \cdots, \varepsilon_{m-2i}$. Therefore its coefficients equal to*

$$-\binom{m}{2i}(2i-1)!! + \sum_{n=1}^{i-1}(-1)^{n-1}\binom{m}{2n}\binom{m-2n}{2i-2n}(2n-1)!!(2i-2n-1)!! \tag{2.8}$$

*If $i$ is odd, then all terms cancel each other except $n = i$, which gives*

$$(-1)^i\binom{m}{2i}(2i-1)!!.$$

*When $i$ is even we have equation 2.8 equals to $m(m-1)\cdots(m-2i+1)$ times*

$$-\frac{(2i-1)!!}{2i!} - \sum_{n=1}^{i-1}(-1)^n\frac{(2n-1)!!(2i-2n-1)!!}{2n!(2i-2n)!}$$

$$= -\sum_{n=1}^{i}\frac{(-1)^n}{2^n n! 2^{i-n}(i-n)!} = -\frac{1}{2^i i!}\sum_{n=1}^{i}\frac{(-1)^n i!}{n!(i-n)!}$$

$$= -\frac{1}{2^i i!}\big((1-1)^i - 1\big) = \frac{1}{2^i i!}.$$

*Therefore the coefficient of $\Delta_{m-2i}$ equals to*

$$\frac{1}{2^i i!}m(m-1)\cdots(m-2i+1) = \binom{m}{2i}(2i-1)!!.$$

Now if we have $\gg \log(1/\delta)\epsilon^{-2}$ samples, using equation 1.5 we have $\Delta_m \ll \epsilon\sigma^m$. Therefore, applying the above lemma we get

$$\varepsilon_m = \hat{P}_m - P_m \ll \epsilon k\sigma^m e^{0.5(\frac{m}{\sigma})^2}. \tag{2.9}$$

We obtain equation 2.9 by considering the Taylor expansion of $e^{0.5(\frac{m}{\sigma})^2}$.

## 3 NEWTON'S IDENTITIES AND ROOTS OF POLYNOMIALS

So far we established that in the case of spherical Gaussian using the moments we can easily calculate $P_m$. Now consider our parameter polynomial:

$$P(x) := \prod_{i=1}^{k}(x - \mu_i) = x^k - e_1 x^{k-1} + \cdots + (-1)^k e_k \tag{3.1}$$

By using Newton's identities we can write coefficients $e_n$, in terms of $P_m$, with $m \leq n$. In general we have

$$e_n = \sum_{j=1}^{n}(-1)^j e_{n-j} P_j. \tag{3.2}$$

Similar to the definition 2.4 we define

**Definition 3** *Let $\hat{e}_0 = 1$ and $\hat{e}_1 = \hat{P}_1$. Recursively we define*

$$\hat{e}_n = \sum_{j=1}^{n} (-1)^j \hat{e}_{n-j} \hat{P}_j. \tag{3.3}$$

Next we estimate how close $\hat{e}_n$ is to $e_n$.

**Lemma 3.1** *Assuming we have $\asymp \log(1/\delta)\epsilon^{-2}$ samples from our mixture, and $P_1 = 0$, and we have that*

$$|\hat{e}_m - e_m| \ll \epsilon K (2\sigma)^m e^{0.5\left(\frac{m}{\sigma}\right)^2} \tag{3.4}$$

**Proof. 3** *Let $E_n = \hat{e}_n - e_n$. Using equation 3.3 and equation 3.2, we have that*

$$E_n \ll \sum_{i=1}^{n} \left( e_{n-i}\varepsilon_i + E_{n-i}P_i + E_{n-i}\varepsilon_i \right) \tag{3.5}$$

*We use following bounds on each of the terms: For $i \geq$ we have $P_m < \sigma^m$. Using equation 2.9 we have $\varepsilon_m \ll \epsilon k \sigma^m e^{0.5(m/\sigma)^2}$. Using the induction hypothesis for $i < n$, $E_{n-i} \ll \epsilon k(2\sigma)^{n-i} e^{0.5(n-i/\sigma)^2}$. From (11) we have $e_n \ll \left(\frac{6e}{n}\right)^{n/2} \sigma^n$. Applying these bound to the first term inside the summation in equation 3.5 we get*

$$\sum_{i=1}^{n} e_{n-i}\varepsilon_i \ll \epsilon \sum_{i=1}^{n} \left(\frac{6e}{n-i}\right)^{(n-i)/2} \sigma^{n-i} \sigma^i e^{0.5(i/\sigma)^2}$$

$$\epsilon k \sigma^n \sum_{i=1}^{n} \left(\frac{6e}{n-i}\right)^{(n-i)/2} e^{0.5(i/\sigma)^2} \ll \epsilon k \sigma^n e^{0.5(n/\sigma)^2}.$$

*As for the term $\sum_{i=1}^{n} E_{n-i}P_i$, using mentioned bounds gives equation 3.4. The third error term is obviously smaller than the other two.*

We prove that in the spherical case having first $k$-moment is enough to uniquely determine parameters of the mixture.

**Lemma 3.2** *Let $G$ be a $k$-mixture of spherical Gaussian (KGMM):*

$$G = \frac{1}{k} \sum_{i=1}^{k} \mathcal{N}(\mu_i, 1).$$

*We have that first $k$ moments of $G$ uniquely determine parameters $\mu_1, \cdots, \mu_k$.*

**Proof. 4** *By using Lemma 2.2 knowing $M_1, \cdots, M_k$, we can uniquely find $p_1, \cdots, p_k$. Assume that $\check{M}_1 \cdots, \check{M}_k$, would also give us $p_1, \cdots, p_k$. Then, using the first equation in equation 1.2 we have $M_1 = \check{M}_1$, by the second equation, we have $M_2 = \check{M}_2$, and so on.*

*Since we have $P_1, \cdots, P_k$, using Newton's identities we can determine coefficients of $P(x)$ in equation 3.1. This polynomial has at most $k$-distinct roots, and we know that $\mu_1, \cdots, \mu_k$ are roots of $P$, therefore first $k$ moments uniquely determine parameters of the mixture.*

## 4 APPROXIMATING COEFFICIENTS USING SAMPLES

So far we showed that we can construct a polynomial whose roots are close to $\mu_1, \cdots, \mu_k$. Define

$$\hat{P}(x) := \sum_{n=0}^{k} \hat{e}_n x^n,$$

where we obtained $\hat{e}_n$ as in equation 3.3. By using Lemma 3.1 we have that $\hat{P}(x)$ is an approximation of $P$. Our task is to measure how close are roots of $\hat{P}$ to the roots of $P$. We apply the following

theorem of Beauzamy (10)

Let $Q(x) = \sum_{i=0}^{k} a_i x^i$ be a polynomial with complex coefficients and degree k. The Bombieri's norm of $Q$ is defined as

$$B(Q) = \Big( \sum_{i=0}^{k} \frac{|a_i|^2}{\binom{k}{i}} \Big)^{1/2}. \tag{4.1}$$

[Beauzamy] Let $P$ and $\hat{P}$ be polynomials of degree $k$. Assume that $B(\hat{P} - P) < \varepsilon$. If $x$ is any zero of $P$ there exist zero $y$ of $\hat{P}$ with

$$|x - y| < \frac{k(1 + |x|^2)^{k/2}}{|\hat{P}'(x)|} \varepsilon. \tag{4.2}$$

If $\varepsilon$ is small enough, namely

$$\varepsilon < \frac{|P'(x)|}{k(1 + |x|^2)^{(k-1)/2}},$$

then

$$|x - y| < \frac{k(1 + |x|^2)^{k/2}}{|P'(x)|} \varepsilon. \tag{4.3}$$

**Proof. 5 (Proof of Theorem 0.3)** *We apply the above theorem to examine how close are roots of $\hat{P}$ to roots of P. Using Lemma 3.1 we know that if we have $\asymp \log(1/\delta)\varepsilon^{-2}$ samples from the mixture, then*

$$B(\hat{P} - P) \ll \varepsilon \Big( \sum_{i=0}^{k} \frac{k^2 e^{(i/\sigma)^2} (2\sigma)^{2i}}{\binom{k}{i}} \Big)^{1/2} \ll \varepsilon k e^{0.5(k/\sigma)^2} (2\sigma)^k.$$

*Therefore by Beauzamy's theorem, if*

$$\varepsilon < \frac{\prod_{j \neq m}(\mu_m - \mu_j)}{k^2 (1 + |\mu_m|^2)^{(k-1)/2} (2\sigma)^k e^{0.5(k/\sigma)^2}}$$

*there exist a root $\hat{\mu}_m$ of $\hat{P}$ such that*

$$|\hat{\mu}_m - \mu_m| < \frac{k^2 (1 + |\mu_m|^2)^{k/2} (2\sigma)^k e^{0.5(k/\sigma)^2}}{\prod_{j \neq m}(\mu_m - \mu_j)} \varepsilon. \tag{4.4}$$

*This gives us the proof of Theorem 0.3.*

**Proof. 6 (Proof of corollary 0.2)** *First note that in the statement of Beauzamy's theorem and the proof of Theorem 0.3, we used $\varepsilon$ to indicate the distance between coefficients of polynomial. Going forward we use $\epsilon$ to indicate the accuracy we expect the statement of the corollary.*

*To get Corollary 0.2, LHS of equation 4.4 must be smaller than the accuracy :*

$$|\hat{\mu}_m - \mu_m| < \epsilon,$$

*for all $m$.*

*In order to have this, $\varepsilon$ in the RHS of equation 4.4, must be smaller than $\epsilon$. Therefore we must have*

$$\varepsilon = \epsilon \frac{\prod_{j \neq m}(\mu_m - \mu_j)}{k^2 (1 + |\mu_m|^2)^{k/2} (2\sigma)^k e^{0.5(k/\sigma)^2}}.$$

*Number of samples we need to get this accuracy is*

$$\gg \log(\delta^{-1})\varepsilon^{-2} = \log(\delta^{-1})\epsilon^{-2} \Big| \frac{\prod_{j \neq m}(\mu_m - \mu_j)}{k^2 (1 + |\mu_m|^2)^{k/2} (2\sigma)^k e^{0.5(k/\sigma)^2}} \Big|^{-2}.$$

*We use*

$$1 + \mu_m^2 \ll 1 + k\sigma^2,$$

*and we set*

$$c(\sigma, k) = \left| k^2 (1 + k\sigma^2)^{k/2} (2\sigma)^k e^{0.5(k/\sigma)^2} \right|^2$$

*Therefore, the sample complexity is*

$$\log(\delta^{-1}) \frac{c(\sigma, k)}{\left| \mathcal{P}_m \right|^2} \epsilon^{-2}. \tag{4.5}$$

*We complete the proof of the corollary by taking the minimum over $m$.*

**Proof. 7 (Proof of corollary 0.1)** *We assumed $\min_{i \neq j} (\mu_i - \mu_j) > \epsilon$, therefore applying Corollary 0.2, we have that $\min_m \mathcal{P}(\mu_m) > \epsilon^{k-1}$. This indicates that the denominator in equation 4.5 is bigger than $\epsilon^{k-1}$. We would like to learn our parameters with $\epsilon/100$-accuracy. Therefore the number of samples we need is*

$$10^4 \log\left(\frac{1}{\delta}\right) \epsilon^{-2k+2} c(\sigma, k) \; \epsilon^{-2}.$$

*This completes the proof.*

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
