**On Gaussian Mixture Models**

## THIS IS A LONGER VERSION OF THE PAPER. IT IS INCLUDED TO HELP WITH REVIEW PROCESS.

N/A

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

There are various ways to solve a system of equations, however the first barrier here is that we can only approximate $M_i$, $1 \leq i \leq 2k$, using the empirical data.

Given $n$ i.i.d. samples $x_1 \cdots x_n$ from our mixture, the $m$-th empirical moment is

(1.3)
$$\hat{M}_m = \frac{1}{n}\sum_{j=1}^{n}x_j^m.$$

Note that $\hat{M}_m$ is random variable with mean $M_i$ and variance $\sigma^{2m}/n$, where $\sigma^2$ is the variance of the $k$-mixture. By Chebyshev's inequality we have

(1.4)
$$\mathbb{P}\Big[|\hat{M}_m - M_m| > r\frac{\sigma^m}{\sqrt{n}}\Big] < \frac{1}{r^2}.$$

Take $n \geq \epsilon^{-2}$, then the portion of samples for which we have

$$|\hat{M}_m - M_m| > r\epsilon\sigma^m,$$

is bounded by $r^{-2}$. This would suffice for our purposes, take for example $r = 10$, then we know that for 99% of samples we have

$$|\hat{M}_m - M_m| < 10\epsilon\sigma^m.$$

This can be improved by taking samples in groups, calculate $\hat{M}_m$ for each group and look at the median of these estimates. In [5][Lemma 3.2] Hardt

and Price proved that given $n \gg \log(1/\delta)\epsilon^{-2}$, samples from the mixture, with probability $1 - \delta$ we have:

(1.5)
$$|\hat{M}_i - M_i| < \epsilon\sigma^i.$$

Going forward we will use (1.5).

## 1.1. System of Equations and Perturbation of Coefficients.

In practice, we often have to approximate these moments. Consequently, a relevant question arises: How robust is the system (1.2) when subjected to perturbations in its constant terms?

**Problem 1.** *Let $\mathcal{M}_r$ be defined as in (1.1), and consider the following system of equations:*

$$M_1 + \varepsilon_1 = \frac{1}{k}\sum_{i=1}^{k}\mathcal{M}_1(\mu_i, \sigma_i)$$

$$M_2 + \varepsilon_2 = \frac{1}{k}\sum_{i=1}^{k}\mathcal{M}_2(\mu_i, \sigma_i)$$

(1.6)
$$\cdots$$

$$M_{2k} + \varepsilon_{2k} = \frac{1}{k}\sum_{i=1}^{k}\mathcal{M}_{2k}(\mu_i, \sigma_i).$$

*We know that for $\varepsilon_1 = \varepsilon_2 = \cdots = \varepsilon_{2k} = 0$, the system has a real solution:*

$$X = (\mu_1, \cdots, \mu_k, \sigma_1, \cdots, \sigma_k).$$

*Now consider the system with perturbation $\varepsilon = (\varepsilon_1, \cdots, \varepsilon_{2k})$ subject to $\varepsilon_i \ll \epsilon\sigma^i$, where $\sigma > 0$ is fixed. Let $X_\varepsilon = (\hat{\mu}_1, \cdots, \hat{\mu}_k, \hat{\sigma}_1, \cdots, \hat{\sigma}_k)$ be a solution to the system with perturbation $\varepsilon$.*

*Is it possible to establish a bound such as*

(1.7)
$$\|X - X_\varepsilon\|_2 \ll C_k\epsilon^{\alpha_k},$$

*where $C_k$ and $\alpha_k$ are constants that depend on $k$?*

We solve this problem in the context of the uniform spherical case. Our approach involves constructing a polynomial whose coefficients closely resemble those of our parameter polynomial. Subsequently, our objective is to demonstrate that the roots of this polynomial provide a robust approximation of the parameters we seek to learn.

In general, roots of polynomials are sensitive to the perturbation of the coefficients. A famous example is Wilkinson's polynomial:

$$\omega(x) = (x - 1)(x - 2)\cdots(x - 20).$$

If the coefficient of $x^{19}$ is decreased from $-210$ by $2^{-23}$ to $-210.0000001192$, then the the root at $x = 20$ grows to $x \simeq 20.8$

## 2. THE UNIFORM SPHERICAL CASE

In this section we prove some necessary lemmas that will help us recover parameters of the mixture. We assume that all variances are equal to 1. This assumption simplifies equation (1.2).

Our final goal is to estimate coefficients of the following polynomial:

$$P(x) := \prod_{i=1}^{k}(x - \mu_i),$$

using information we get from (1.2).

Define

$$P_m(\mu_1, \cdots, \mu_k) = \mu_1^m + \cdots + \mu_k^m.$$

We can write each moments in (1.2), in terms of $P_m$ and vice versa. For example

$$M_1 = \frac{1}{k}P_1(\mu_1, \cdots, \mu_k)$$

$$M_2 = 1 + \frac{1}{k}P_2(\mu_1, \cdots, \mu_k)$$

(2.1)                                                       $\cdots$

$$M_m = \frac{1}{k}\sum_{j=0}^{\lfloor m/2 \rfloor} c_{m,j}P_{m-2j}.$$

Using (2.1), and given $M_i$ for $i \leq m$, recursively we can find $P_i$ for $i \leq m$, in terms of $M_i$. For example

$$P_1 = kM_1$$

(2.2)                    $$P_2 = kM_2 - k$$

$$P_3 = kM_3 - 3kM_1 \cdots.$$

Let us precisely calculate $c_{m,j}$ first, then we move to approximate $P_m$ using imperial moments we obtain in (1.3).

**Lemma 2.1.** *For $m \geq 1$ we have that*

(2.3)          $$\mathcal{M}_m(\mu, 1) = \mu^m + \sum_{i=1}^{m/2}\binom{m}{2i}\mu^{m-2i}(2i-1)!!,$$

*where $(2i-1)!!$ is the product of odd numbers less than $2i-1$.*

**Remark 1.** Using the above lemma we immediately have the expansion of $M_m$ in terms of $P_i$ :

(2.4)          $$kM_m = P_m + \sum_{i=1}^{m/2}\binom{m}{2i}(2i-1)!!P_{m-2i}.$$

In the next lemma we show that

$$(2.5) \qquad P_m = kM_m + k \sum_{i=1}^{m/2} (-1)^i \binom{m}{2i} (2i-1)!! M_{m-2i}.$$

*Proof of Lemma 2.1.* We have

$$\mathcal{M}_m(\mu, 1) = \frac{1}{\sqrt{2\pi}} \int x^r e^{-\frac{1}{2}(x-\mu)^2}$$

$$= \frac{1}{\sqrt{2\pi}} \int (x+\mu)^m e^{-\frac{1}{2}x^2} = \sum_i \binom{m}{i} \mu^{m-i} E(x^i),$$

where $x \sim \mathcal{N}(0,1)$.

We use $E(zf(z)) = E(f'(z))$, therefore we have

$$E(x^m) = E(xx^{m-1}) = (m-1)E(x^{m-2}).$$

Also note that $E(x^m)$ is zero if $m$ is odd. This will give the lemma. $\qquad \square$

We now define a new object that is a empirical approximation of $P_m$.

**Definition 2.** *Let $\hat{M}_m$ be as (1.3), following (2.5), we define*

$$(2.6) \qquad \hat{P}_m = k\hat{M}_m + k \sum_{i=1}^{m/2} (-1)^i \binom{m}{2i} (2i-1)!! \hat{M}_{m-2i}.$$

**Lemma 2.2.** *We have that $P_m$ satisfy:*

$$(2.7) \qquad P_m = kM_m + k \sum_{i=1}^{m/2} (-1)^i \binom{m}{2i} (2i-1)!! M_{m-2i}.$$

*Moreover, Let $\varepsilon_m = \hat{P}_m - P_m$ and $\Delta_m = \hat{M}_m - M_m$. We have that*

$$(2.8) \qquad \varepsilon_m = k\Delta_m + k \sum_{i=1}^{m/2} (-1)^i \binom{m}{2i} (2i-1)!! \Delta_{m-2i}.$$

*Proof.* We proceed with using induction, we have that $kM_1 = P_1(\mu_1, \cdots, \mu_k)$ and $kM_2 = 1 + P_2(\mu_1, \cdots, \mu_k)$. This gives us (2.7) and (2.8) for the base cases. We assume (2.8) for $m-2, m-4, \cdots$ and we prove it for $m$.

Using (2.4), we have the following recursive identity relating $\varepsilon_m$ to $\Delta_m$ and $\varepsilon_{m-2}, \varepsilon_{m-4}, \cdots$.

$$(2.9) \qquad \varepsilon_m = k\Delta_m - \sum_{i=1}^{m/2} \binom{m}{2i} (2i-1)!! \varepsilon_{m-2i}.$$

We use our induction hypothesis in (2.9) and we get everything in terms of $\Delta_m, \Delta_{m-2}, \cdots$. Only thing remaining is to calculate the coefficients of $\Delta_{m-2i}$ for $0 < i \leq m/2$.

The term $\Delta_{m-2i}$ appears in (2.9) expansion of $\varepsilon_m, \varepsilon_{m-2}, \cdots, \varepsilon_{m-2i}$. Therefore its coefficients equal to

$$(2.10) \quad -\binom{m}{2i}(2i-1)!! + \sum_{n=1}^{i-1}(-1)^{n-1}\binom{m}{2n}\binom{m-2n}{2i-2n}(2n-1)!!(2i-2n-1)!!$$

If $i$ is odd, then all terms cancel each other except $n = i$, which gives

$$(-1)^i\binom{m}{2i}(2i-1)!!.$$

When $i$ is even we have (2.10) equals to $m(m-1)\cdots(m-2i+1)$ times

$$-\frac{(2i-1)!!}{2i!} - \sum_{n=1}^{i-1}(-1)^n\frac{(2n-1)!!(2i-2n-1)!!}{2n!(2i-2n)!}$$
$$= -\sum_{n=1}^{i}\frac{(-1)^n}{2^n n! 2^{i-n}(i-n)!} = -\frac{1}{2^i i!}\sum_{n=1}^{i}\frac{(-1)^n i!}{n!(i-n)!}$$
$$= -\frac{1}{2^i i!}\left((1-1)^i - 1\right) = \frac{1}{2^i i!}.$$

Therefore the coefficient of $\Delta_{m-2i}$ equals to

$$\frac{1}{2^i i!}m(m-1)\cdots(m-2i+1) = \binom{m}{2i}(2i-1)!!.$$

$\square$

Now if we have $\gg \log(1/\delta)\epsilon^{-2}$ samples, using (1.5) we have that

$$\Delta_m \ll \epsilon\sigma^m.$$

Therefore, Using the above lemma we get

$$(2.11) \qquad \varepsilon_m = \hat{P}_m - P_m \ll k\sigma^m e^{0.5(\frac{m}{\sigma})^2}.$$

We obtain (2.11) by considering the Taylor expansion of $e^{0.5(\frac{m}{\sigma})^2}$ and comparing it to (2.8).

## 3. Newton's identities and roots of polynomials

So far we established that in the case of spherical Gaussian using the method of moments we can easily calculate $P_m$. Now consider the following polynomial:

$$(3.1) \qquad P(x) := \prod_{i=1}^{k}(x - \mu_i) = x^k - e_1 x^{k-1} + \cdots + (-1)^k e_k$$

Recall that $P$ is our parameter polynomial. By using Newton's identities we can write coefficients $e_n$, in terms of $P_m$, with $m \leq n$. For example we

have

$$e_1 = P_1$$
$$2e_2 = P_1^2 - P_2$$
$$3e_3 = \tfrac{1}{2}P_1^3 - \tfrac{3}{2}P_1P_2 + P_3, \cdots$$

In general we have

(3.2)
$$e_n = \sum_{j=1}^{n} (-1)^j e_{n-j} P_j.$$

Similar to the definition 2.6 we define

**Definition 3.** *Let $\hat{e}_0 = 1$ and $\hat{e}_1 = \hat{P}_1$. Recursively we define*

(3.3)
$$\hat{e}_n = \sum_{j=1}^{n} (-1)^j \hat{e}_{n-j} \hat{P}_j.$$

By this definition and (1.5) and by some calculation we obtain

$$\hat{e}_1 = O(\epsilon k \sigma)$$
$$2\hat{e}_2 = -k(\sigma^2 - 1) + O(\epsilon k \sigma^2)$$
$$3\hat{e}_3 = KM_3 + O(\epsilon k^2 (\sigma^3 + 1)).$$

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

**Description of the algorithm.** Using samples from the mixture we we easily compute $\hat{M}_1, \cdots, \hat{M}_k$ by (1.3).

We use (2.6) and from $\hat{M}_1, \cdots, \hat{M}_k$ we recover $\hat{P}_1, \cdots, \hat{P}_k$.

Next, by using (3.4) from $\hat{P}_1, \cdots, \hat{P}_k$, recursively we obtain $\hat{e}_1, \cdots, \hat{e}_k$. At this point we have coefficients of our empirical parameters polynomial.

Finally we use one of algorithms for finding roots of univariant polynomials. For example one can use the Aberth method [1]. Theorem 0.3 gives us a guarantee in terms of how close are roots of the empirical parameters polynomial to parameters of the mixture.