# OpenReview forum: "On Gaussian Mixture Models"
_ICLR.cc/2024/Conference — Submitted to ICLR 2024_

### Official Review · Reviewer_WTMq · 2023-10-27

**Soundness:** 3 good
**Presentation:** 2 fair
**Contribution:** 3 good
**Rating:** 6
**Confidence:** 3

**Summary:**

The paper studies estimating the mean parameters of a equi-weighted mixture of $k$ Gaussians with unit variance: $\Gamma = \frac{1}{k} \sum_{i=1}^k \mathcal{N}(\mu_i, 1)$. The paper gives an upper bound on the sample complexity to estimate the mean parameters within an error of $\epsilon$. The results are interesting and novel. The paper can be improved by correcting some mistakes and fixing several typos.

Notation:
* $\mu_1, \dots, \mu_k$ are the means to be estimated
* $M_m$: m'th moment of the mixture
* $P_m = \sum_{i=1}^m \mu_i^m$
* $e_m$: elementary symmetric polynomials of mean parameters
* $P(x) = \Pi_{i=1}^k (x-\mu_i)$ is a polynomial whose roots are the mean paramters

Some key points
* We can write $P_m = f(M_1, \dots, M_m)$ for some function f
* We can write $e_m = g(P_1, \dots, P_m)$ for some function g

How to estimate $\mu_1, \dots, \mu_k$ from a sample drawn from $\Gamma$?
* Compute empirical moments $\hat M_m$
* Plugin estimate $\hat P_m = f(\hat M_1, \dots, \hat M_m)$
* Plugin estimate $\hat e_m = f(\hat P_1, \dots, \hat P_m)$
* Form the polynomial $\hat P(x) = \sum_{i=1} \hat e_i x^i$.
* The roots of this polynomial yield estimates of the means

The authors show that if the means are well-separated, then given $\log(1/\delta) \epsilon^{-2}$ samples, with probability $1 -\delta$, $|\hat \mu_i - \mu_i | \leq c \epsilon$ where $c$ depends on $k, var(\Gamma)$ and a mean-separation metrics $\Pi_{i \neq m} (\mu_m - \mu_i), m \in [k]$. On a high-level, the proof
* ties together $\hat M$ and $M$ (previously known)
* then bounds $\hat P_m - P_m$
* then bounds $\hat e - e$
* then gives root difference bounds between $\hat P(x), P(x)$ from real analysis literature

**Strengths:**

The results are interesting and novel.

**Weaknesses:**

There are several mistakes and typos which make this theoretical paper hard to read.

**Questions:**

Comments/questions/typos

* Update the title, currently it is a placeholder from ICLR formatting instructions.
* page 2 second line: Why is it $3k-2$ parameters? I understand $3k-1$ (due to the fact that weights need to sum up to 1).
* What Corollary 0.1 a corollary to? Rename it to a Proposition?
* Sentence below equation (0.2): make it tighter, maybe $\Omega(\epsilon^\ell)$.
* Theorem 0.3: first line: mean does not seem to be 0? There are some other places as well where the mean is stated to be 0.
* Equation (1.2): $\mathcal{M}_d$ should be $\mathcal{M}_m$?
* Sentence below eq (1.3): The variance of $x^m$ is not $\sigma^{2m}$. You seem be missing a constant?
* Senetence above Lemma 2.1: imperial → empirical
* Lemma 2.1 last line: less than "or equal to" $2i-1$.
* Eq (2.9): maybe write one/two steps more to explain it.
* Eq (3.2): left hand side should have a factor of $n$. Right hand side should have $j-1$ in the exponent, not $j$. Also say that $e_0 = 1$. Ditto for Definition 3.
* I believe, some constants in the proofs need to be changed because of the above mistake.
* Lemma 3.1: The statement is missing "with probability at least $1-\delta$".
* Eq (3.4): Is the $K$ used here defined before?
* Line below eq (3.5): $i\geq m$?
* Proof 3: Second display, last inequality: explain how to derive this. I am not sure if this is true for small $n$.
* Proof 4.: In the first line, $p$ should be $P$.
* Eq (4.2): How to handle the case where the denominator is $0$?
* Display above eq (4.4): How is $|\hat P'(x)$ bounded by $\Pi_{j\neq m} (\mu_m - \mu_j)$ here?
* There are duplicates in References.

I have not checked a few proofs: Lemma 2.2, Proof 3.

---

> ### Author Response · Authors · 2023-11-21
> **Response to Reviewer WTMq**
>
> Thank you for your thorough review of my paper; I greatly appreciate your careful reading. I would also like to draw your attention to my response to reviewer EBv4. The reviewer mentioned a paper [1] that I was not aware of before submission. Upon comparing my results with those presented in [1], I believe it demonstrates the advantages of my findings.
>
> [1] Y. Wu. Optimal estimation of Gaussian mixtures via denoised method of moments. The Annals of Statistics, 48:1987–2007, 2020.
>
> $\textbf{Reviewer's comments.}$ Weaknesses:
> There are several mistakes and typos which make this theoretical paper hard to read.
>
> $\textbf{Response:}$ I am sorry for the problem with the title and the writing. I explained the circumstance in my general response to reviewers.
>
> Questions:
> $\textbf{Comments/questions/typos}$
>
> Q1- Update the title, currently it is a placeholder from ICLR formatting instructions.
>
> A1- Sure.
>
> Q2- page 2 second line: Why is it 3k-2 parameters? I understand (due to the fact that weights need to sum up to 1).
>
> A2- You are correct, I need to clarify this, thanks for noticing. They assume a sort of separation between parameters, here is a quote from their paper, the paragraph before theorem 2.1: " the mixtures differ in at least one of $\mu_i$ and $\sigma_i$, so the lower bound now only applies to algorithms that recover both  $\mu_i$ and $\sigma_i$  do the desired precision." So they assume there is a separation between 2 of the parameters, and therefore the sample complexity is lower than the worst case scenario.
>
> Q3- What Corollary 0.1 a corollary to? Rename it to a Proposition?
>
> A3- Will fix it.
>
> Q4- Sentence below equation (0.2): make it tighter, maybe $\Omega(\epsilon^l).$
>
> A4- Done.
>
> Q5- Theorem 0.3: first line: mean does not seem to be 0? There are some other places as well where the mean is stated to be 0.
>
> A5- Will fix it, thanks.
>
> Q6- Equation (1.2):  M_d  should be M_m?
>
> A6- Fixed.
>
> Q7- Sentence below eq (1.3): The variance of  $x^m$ is not $\sigma^{2m}$ . You seem to be missing a constant?
>
> A7- Do you mean that the variance of $1/n \sum (x_i)^m$ is a constant time $\sigma^{2m}/n?$
>
> Q8- Senetence above Lemma 2.1: imperial → empirical
>
> A8- Fixed.
>
> Q9- Lemma 2.1 last line: less than "or equal to"
>
> A9- Fixed.
>
> Q10- Eq (2.9): maybe write one/two steps more to explain it.
>
> A10- Sure.
>
> Q11- Eq (3.2): left hand side should have a factor of $n$ . Right hand side should have $j-1$ in the exponent, not $j$. Also say that $e_0=1$. Ditto for Definition 3.
>
> A11- Done.
>
> Q12- I believe some constants in the proofs need to be changed because of the above mistake.
>
> A12- A factor of $1/n$ comes into the RHS of (3.2). This will help, I will clarify this.
>
> Q13- Lemma 3.1: The statement is missing "with probability at least ".
>
> A13- Fixed, Thanks.
>
> Q14- Eq (3.4): Is the $K$  used here defined before?
>
> A14- Sorry, this is a typo. Should be $k$.
>
> Q15- Line below eq (3.5): $i \geq m?$
>
> A15- There were two typos, so I fixed them. Thanks for reading the manuscript carefully.
>
> Q16- Proof 3: Second display, last inequality: explain how to derive this. I am not sure if this is true for small n.
>
> A16- You are correct that a constant that depends only on 6e = 16.3..., pops out on the RHS. That is why I used the $\ll$ notation. I will add a line to clarify this.
>
> Q17- Proof 4.: In the first line, p should be P.
>
> A17- Fixed.
>
> Q18- Eq (4.2): How to handle the case where the denominator is 0?
>
> A18- Beauzam made a comment in a remark following the theorem. When the denominator is 0, the theorem becomes void, essentially stating that $∣x−y∣<\infty$, which is a trivial assertion. This situation arises when we have a double root, implying that two of the means are equal. However, if two of the means are equal, they effectively constitute the same Gaussian component, given our assumption that variances are both equal to one. In essence, we have $N(\mu, 1)+ N(\mu, 1)= 2N(\mu, 1).$
>
> Q19- Display above eq (4.4): How is  $P'(x)$   bounded by $\prod(\mu_m-\mu_j)$ here?
>
> A19- Sure.
>
> Q20-There are duplicates in References.
>
> A20-Fixed.

---

> > ### Comment · Reviewer_WTMq · 2023-11-22
> > **Response to author**
> >
> > Thanks for addressing my questions.
> >
> > Q7: Yes.
> >
> > Q18: Q18- Eq (4.2): How to handle the case where the denominator is 0?
> >
> > A18- Beauzam made a comment in a remark following the theorem. When the denominator is 0, the theorem becomes void, essentially stating that $|x-y| < \infty$, which is a trivial assertion. This situation arises when we have a double root, implying that two of the means are equal. However, if two of the means are equal, they effectively constitute the same Gaussian component, given our assumption that variances are both equal to one.
> >
> > Q18-followup: Even with you explanation, it is not clear how your proof argument flows in the case where the denominator is 0. How does (4.3) follow from (4.2)? Also, the double roots are for $\hat{P}'$, not $P$.

---

> > > ### Author Response · Authors · 2023-11-23
> > > **Response to reviewer WTMq**
> > >
> > > The claim is not that 4.3 follows from 4.2. They are separate parts of Beazamy's theorem. In 4.2, he states that if the Bombieri norm of two polynomials is small, then we have the bound 4.2, for the distance between their roots. In 4.3, he states that if the Bombieri norm is smaller than a certain threshold (as indicated by the inequality before 4.3), then we have the bound expressed in 4.3. I used 4.3 in my proof. That's why I only needed to consider the roots of the parameter polynomial P, and in that context, I can assume that we do not have double roots. I hope this clarifies the issue.

---

### Official Review · Reviewer_EBv4 · 2023-10-29

**Soundness:** 4 excellent
**Presentation:** 2 fair
**Contribution:** 1 poor
**Rating:** 3
**Confidence:** 5

**Summary:**

The paper studies the problem of sample complexity for learning mixtures of $k$ univariate, equal-weighted, standard Gaussians.
The authors devised an algorithm that takes $n$ samples from such a mixture and returns $k$ estimated means,
They showed that if $n > \frac{C}{\epsilon^2}$ where $C$ is a factor depending on the separation of means, all estimated means have an additive error of $\epsilon$.
Note that when the separations of the means are constant, the factor $C$ is a constant.
When the separations of the means are $\sim \epsilon$, the factor can be $\frac{1}{\epsilon^{2k-2}}$ and hence the final sample bound is $\frac{1}{\epsilon^{2k}}$.
The main idea of the proof is moment based.
The authors consider the polynomial whose roots are the means of the mixture and expand the coefficients in terms of the moments by Hermite polynomials.
Then, one can use samples to estimate the coefficients and solve the polynomial with the estimated coefficients.
Hence, the roots are supposed to be close to the true roots which are the means.

**Strengths:**

- The result achieves the optimal rate and explains the connection between the sample complexity and the separations of the means.

**Weaknesses:**

- The proofs are mostly calculations.
I believe most of the lemmas are not fundamentally novel or can be easily follow from previously known results.
This paper appears to primarily reiterate known results and may be somewhat remote from making a substantial contribution.
For example: Aren't Lemma 2.1 and 2.2 the Hermite polynomial expansion?
The authors may also want to cite the paper "Optimal estimation of Gaussian mixtures via denoised method of moments" by Yihong Wu, Pengkun Yang.
Even though the problem definitions are not exactly the same, there is discernible overlap in the techniques employed to address the underlying challenges.


- The paper may need some work to polish it.
For example:
Make the title a bit more informative, start the section number from 1 instead of 0 and other numbering systems...

**Questions:**

Notes:
- In (0.1): I think $\omega_i$ also needs to be $\geq 0$.
- In (1.2): The authors may want to introduce Hermite polynomials before (1.2). Also, should $\mathcal{M}_d$ be $\mathcal{M}_m$?
- References: [6] and [7] are the same.

---

> ### Author Response · Authors · 2023-11-21
> **Response to Reviewer EBv4**
>
> Thank you for taking the time to review my paper and for making reference to the work of Wu and Yang. I will compare my results with theirs and I believe this would clarify the contribution.
>
> In Theorem 2, Wu and Yang consider scenarios in which a mixture has $k_0$ separated clusters. To underscore the advantage of my own results, assume $k_0 =2$,  and we have 2 clusters separated by a difference of 1 + O(ε). For instance, the first cluster possesses means of {-mε, -(m-1)ε, ..., 0, ε, ..., mε}, while the second cluster contains {1-mε, 1-(m-1)ε, ..., 1, 1+ε, ..., 1+ mε}, with k = 4m. Consequently, each cluster consists of k/2 members.
>
> Applying Wu and Yang's theorem 2, we observe that the RHS of equation (11) yields n^(1/(4k-6)). Therefore we need n > (1/ε)^(4k-6) to have the RHS of (11) proportional to ε. This implies that they achieve a saving of ε^(-4) compared to their bound in (8) in the worst-case scenario. In contrast, my Corollary 2 results in a saving of ε^(-k/2) in this case. There is a substantial distinction between saving ε^k and ε^4. This clearly underscores the advantage of my results.
>
> The method of moments leads to a system of equations. Given that we employ samples to approximate moments, it is necessary to consider perturbations in the coefficients of the system. Hence, it is unsurprising that there are similarities between my paper and Wu and Yang's work, including the presence of certain lemmas. However, the divergence lies in how we approach and resolve the system and evaluate the impact of perturbations. In my case, I employ Beauzamy's theorem from mathematical analysis to address this. This is a sophisticated theorem, and applying it provides the distinct advantage between our methods.
>
> Second part of comparison: It is well known that parameter estimation is a more challenging task than returning a distribution that closely matches the mixture in a certain norm.  In their work Wu and Yang return a distribution that is close to the target in Wasserstein distance. In my work, I directly estimate parameters. In Lemma 1, Wu and Yang explore the conditions under which one can transition from the Wasserstein distance to parameter estimation. To elaborate, let's denote the minimum distance between the means as "min," with all weights equal to 1/k. To transfer their bound on the Wasserstein distance to a bound on parameter estimation, they require more than (k/(min*ε))^(4k) samples. Hence, the inverse of the minimum distance to the power of 4k contributes to sample complexity. In contrast, my method only demands the inverse of the minimum distance to rise to the power of the number of consecutive gaps with a minimum distance, which in most cases is much smaller. Therefore they have to use excessive amounts of samples (unnecessary in almost all cases) to get the Wasserstein distance small enough to be able to apply their method in parameter estimation.
>
> The difference between our methods is particularly significant in practice. Consider situations where the distribution of spacing between means follows a Poisson distribution or is GUE-like in the context of the distribution of eigenvalues of large Hermitian matrices. While small gaps are still present, their probability is exceedingly low. The mere possibility of such cases drastically increases the sample complexity when applying their method, whereas my approach only amplifies the sample complexity by (1/min)^(4k * probability of small consecutive gaps), representing a substantial practical advantage.
>
> Moving on to the third point of comparison, a crucial feature of Hardt and Price's (HP) work is their theorem 3.10, which demonstrates that sample complexity depends on the differences between two means. If the gap between means is not small, the sample complexity remains moderate.  Thus, any extension of their result should uphold this characteristic. I accomplish this by introducing the pair correlation function between the means. In their paper, Yihong Wu and Pengkun Yang also make reference to HP's work: "By carefully analyzing Pearson’s method of moments equations, HP showed that the optimal rate for two-component location-scale mixtures is Θ(n^(-1/12)); however, this approach is difficult to generalize to higher order mixtures." As they acknowledge, generalizing HP's approach is challenging, which is precisely the task I pursued in my paper. Initially, I had no idea how I should generalize the feature. However, the natural application of Beauzamy's theorem addressed this challenge.
>
> In conclusion, our bound only matches in the worst case scenario, in any other cases my bound has a clear advantage. My method estimates parameters directly, which is a harder task. Finally, in my opinion, my paper is making a significant contribution.
>
> Overall your comments have been very useful to me. By comparing the paper you referenced with my results and providing the explanations outlined above, I believe my paper has been substantially improved.

---

> > ### Comment · Reviewer_EBv4 · 2023-11-22
> >
> > Thanks for the response.
> >
> > In Wu and Yang's paper, they considered a more general setup that the weights are unknown and hence if we directly apply their result it is straightforward to conclude that the sample complexity in the authors' result is better. However, it is still not clear to me that if an easy modification from Wu and Yang's proof can lead to the authors' result by setting all weights to be $\frac{1}{k}$ even though Wu and Yang made the comment mentioned in this third point of the authors' response. In the second point in the authors' response, the output returned in Wu and Yang's result is the mixing distribution which is defined by the weights and the means. Namely, their result is also parameter estimation though the formal conversion between the parameter estimation and the Wasserstein distance may require extra assumptions. I am not entirely sure this part of the response. I do acknowledge that, as I mentioned in the strengths, the connection between the separation of the means and the samples complexity is a novelty of this paper.
> >
> > In any case, a thorough discussion between the authors' result and Wu and Yang's result is needed. Together with other writing issues, I will keep my score.

---

### Official Review · Reviewer_xe5T · 2023-10-29

**Soundness:** 2 fair
**Presentation:** 1 poor
**Contribution:** 2 fair
**Rating:** 3
**Confidence:** 4

**Summary:**

In this paper, the authors aim to establish the sample complexity of $k$-component Gaussian mixture models (GMMs), that is, to find the optimal number of samples necessary to achieve mixture parameter estimation within some tolerance $\epsilon$ from the true parameters. By using the method of moments, they figure out that a relationship between the sample complexity of one-dimensional GMMs with mean zero and variance $\sigma^2$ and the distribution of spacings among their means. Via that relationship, the author arrives at the sample complexity of $\epsilon^{-2}$.

**Strengths:**

1. Originality: The sample complexity of $k$-component Gaussian mixture models is novel but limited to one-dimensional setting.

2. Quality: All results in the paper are associated with theoretical guarantee. The authors intepret their results well.

**Weaknesses:**

1. Clarity: the presentation of the paper is poor due to the following reasons:
- The title of the paper is missing.
- The abstract does not summarize the paper well. In particular, the authors should highlight that the sample complexity is established in one-dimensional setting.
- At the beginning of the paper, it is too general to say that GMMs have been extensively studied in the field of machine learning. The authors should specify the applications of GMMs in that fields, and cite relevant papers.
- There are some undefined notations, e.g. $\sigma_i$ in the contribution paragraph.
- The corollaries are presented without any theorems introduced before, which makes no sense.
- In the main text, I suggest that the authors should only provide a proof sketch in a separate section and then leave the full proofs in the supplementary material. This would help improve the cohesiveness of the paper.

2. The sample complexity of GMMs derived in the paper is limited to one-dimensional setting. Moreover, the variance of each Gaussian distribution is assumed to be one, which is quite restricted.

3. When discussing about the parameter estimation of GMMs, the authors should involve more relevant papers, not only those using the method of moments but also other methods like Maximum Likelihood, namely [1], [2], [3], [4] and [5].

4. The paper lacks a simulation study to empirically verify their theoretical results. I believe that such numerical experiments would significantly strengthen the paper.

5. The authors should include a discussion paragraph at the end to conclude the paper, discuss about the limittaions of their current technique, and its ability to extend to more general settings of GMMs.

**References**

[1] H. Nguyen. Demystifying Softmax Gating Function in Gaussian Mixture of Experts. In NeurIPS, 2023.

[2] N. Doss. Optimal estimation of high-dimensional Gaussian location mixtures. The Annals of Statistics, 51(1):62 – 95, 2023.

[3] H. Nguyen. Towards Convergence Rates for Parameter Estimation in Gaussian-gated Mixture of Experts. In arXiv, 2023.

[4] Y. Wu. Optimal estimation of Gaussian mixtures via denoised method of moments. The Annals of Statistics, 48:1987–2007, 2020.

[5] H. Nguyen. Statistical Perspective of Top-K Sparse Softmax Gating Mixture of Experts. In arXiv, 2023.

**Questions:**

1. In Corollary 0.1, is the variance $\sigma^2$ of the whole mixture or an individual Gaussian distribution?

2. In Section 0.1, the authors should define the notation $\Omega(\sigma)$, and do not assume that readers will understand.

3. In the Gaussian mixture models, when the mixture weights are generalized to depend on the covariates $X$, namely softmax weights (see [1]), can the current techniques be applied to those settings?

4. In Theorem 0.3, is the mean zero and variance one assumptions for simplicity or necessary for the proof to hold true?

5. What are the main challenges of deriving the sample complexity of GMMs in high-dimensional settings?

6. In equation (4.2), Is the term $\hat{P}'(x)$ always different from zero?

7. The 'Corollary' should be renamed as 'Proposition'.

8. In equation (0.1), the weights $\omega$ need to be non-negative.

**References**

[1] H. Nguyen. Demystifying Softmax Gating Function in Gaussian Mixture of Experts. In NeurIPS, 2023.

---

> ### Author Response · Authors · 2023-11-21
> **Response to Reviewer xe5T**
>
> Thank you for taking the time to review my paper. I will respond to your comments one by one.
>
> Please note that in the spherical setting  d-dimensional can be easily transferred to 1-dimensional. In every dimension you have a mean and the variance is equal to 1.  Hence, d dimensional case transfer to d instances of 1-dimensional case.
>
>
> $\textbf{Reviewer's comment:}$ Weaknesses:
> I will enumerate comments by C1, C2, ...
>
> C1-"Clarity:"
>
> R1- I will respond to each bullet point, with B1, B2, ...
>
> B1-I am sorry for the problem with the title and the writing. I explained the circumstance in my general response to reviewers.
>
> B2-I will add more to the abstract.
>
> B3-I will add applications. I briefly stated that this was originated by Pearson. For example I can add the application that Pearson had in mind, where he analyzed a population of crabs and found that a mixture of two gaussians explained the size of the crab's foreheads.
>
> B4-Fixed.
>
> B5-Since the theorem is more abstract, I put corollaries that are easier to understand first. I can bring the theorem first then corollaries on the referee's recommendation.
>
> B6- I will apply your suggestion.
>
> C2-"The sample complexity..."
>
> R2- I already mentioned that d-dimensional easily transfers to d instances of one-dimensional cases. Regarding the equality of the variance, please note that when we consider a scenario where all variances are equal, it represents the most challenging case for our problem. Essentially, when the set of means is fixed, if all variances are equal, we encounter the worst sample complexity. Therefore, the sample complexity we derive serves as an upper bound applicable to a broader context where variances are not necessarily equal. It's also important to mention that Hardt and Price [1] have previously demonstrated that for 2-mixtures, when means are in close proximity, differences in variances would reduce the sample complexity.
>
> My results can be particularly valuable to researchers and practitioners dealing with the parameter estimation of GMMs. Especially when there is some prior information available about the distribution of spacings between means. I have elaborated further on this in my responses to your questions and in the reply to reviewer EBv4.
>
> As for assuming the mean of the mixture is zero, we can always center our data by subtracting the empirical mean from each data point. This adjustment allows us to work in a setting where the overall mean is zero.
>
> I would also like to draw your attention to two recent papers [2,3] that delve into the spherical case, assuming an additional condition of means being separated by (log k). These papers explore even more specific instances of the problem I studied and have been published in respected ML journals. This underscores that this is an active area of research in the theoretical ML. Nevertheless, I acknowledge that one limitation of my work is the assumption of equal variances.
>
> [2] Mingda Qiao. "A Fourier Approach to Mixture Learning." NeurIPS 2022.
>
> [3] Allen Liu and Jerry Li. 2022. "Clustering mixtures with almost optimal separation in polynomial time."  (STOC 2022).
>
> C3-"When discussing ..."
>
> R3- Thanks for mentioning these papers. I will cite them. Please note that in response to Reviewer EBv4 I extensively compared my result to the one in reference [4].
>
> C4- "The paper lacks a simulation study..."
>
> R4- I wrote Python code to evaluate my algorithm's performance. I generated samples from a mixture of 3 Gaussians (k=3), and the algorithm performed well.  If you mean that I find a data set of a mixture of spherical Gaussians and try the algorithm on it; this is an interesting suggestion, and I will look into it.
>
> C5-"The authors should..."
>
> R5- Sure.
>
>
>
> $\textbf{Questions:}$
>
> I will answer questions one by one, enumerated by A1, A2, ...
>
> A1-it's the variance of the whole mixture, I will clarify this.
>
> A2- Sure.
>
> A3-I have to look more into this question. However, If using empirical moments of samples we can get information on moments of parameters you want to estimate, then using Newton's identities and my method you would be able to get some estimate on the number of samples you need.
>
> A4-The mean zero is for simplicity, however the condition variance equal one is necessary. In a subsequent paper, I will delve into cases where variances may not necessarily equal one, but it's worth noting that this will involve more complex mathematics.
>
> A5- High dimensional setting is more or less straightforward, there are many tricks developed by researchers to reduce the d-dimensional to 1-dimensional. For example see [1], page 8, section on Dimension Reduction.
>
> A6- If it's zero then the theorem is void. Basically says that $|x-y|< \infty$. Beauzam [2] addressed this in comments after the theorem.
>
> A7- I will adrees this.
>
> A8-Sure.
>
> [1] Moritz Hardt, Eric Price,  https://doi.org/10.48550/arXiv.1404.4997
>
> [2] ] Bernard Beauzamy, A Local Quantitative Result. Canad. Math. Bull. Vol. 4

---

> > ### Comment · Reviewer_xe5T · 2023-11-21
> >
> > Dear Authors,
> >
> > Thanks for your response. Since the paper still has much room for improvement, I think the current version does not clear the bar of ICLR, and more importantly, is not ready to be published. Therefore, I decide to keep my score unchanged.
> >
> > Best,
> >
> > Reviewer xe5T

---

### Official Review · Reviewer_Ywzd · 2023-10-30

**Soundness:** 2 fair
**Presentation:** 1 poor
**Contribution:** 1 poor
**Rating:** 3
**Confidence:** 3

**Summary:**

In this paper, the authors look at the problem of learning mixture of Gaussians. There are two points of view for studying this problem. The first is to learn the mixture in total variation distance. The second one which is taken in this paper is to learn the mixture probabilities, means, and covariance matrices up to certain desired precision. The authors claims to give better than existing bounds for this parameter estimation problem. I found the paper is extremely ill-presented and very confusing to read.

The main contribution seems to be an algorithm for the uniform spherical case, where each mixture probability is the same and the standard deviation is 1.

**Strengths:**

I think the problem is well-studied and important in the community.

**Weaknesses:**

- The results are described in a very confusing manner all over the paper. To quote: "... assume that we have gaps of length $\varepsilon$ between consecutive means in our mixture. However, these gaps are isolated, meaning that if $\mu_{n+1}-\mu_n=\varepsilon$, then the
adjacent gaps are significantly larger:  $\mu_{n+2}-\mu_{n+1}>>1$, and  $\mu_{n}-\mu_{n-1}>>1$". What does this mean? On the one hand, you are saying consecutive gaps are small. But immediately after that you are saying certain gaps are significantly large.

-Are you considering the univariate or the multivariate Gaussian case? In equation 0.1 you are saying covariance matrix whereas in the our contribution paragraph, you are saying $\sigma_i=1$ which corresponds to the univariate case. This is very confusing to read.

- Furthermore, the main contribution seems to be an algorithm for the uniform mixture of spherical Gaussians, which seems rather limited and incremental. Also is it already covered by the prior work? See my comment for Questions.

- Now I'll come to the *main weakness* of the paper. The presentation is extremely poor and unprofessional. For example the paper don't even have a title and section title for the starting section. One cannot simply expect reviewers to spend time on a paper where the authors themselves have not spend much time.

**Questions:**

You are saying the main contribution is an algorithm for the case $\omega_i = \frac{1}{k}$ and $\sigma_i = 1$. I though Hard and Price already show that the tight sample complexity is $\Theta(\Sigma^{-12})$ which in this case would be $\Theta(1)$.

Considering the above fact, why is your result interesting? What am I missing here?

**Details Of Ethics Concerns:**

None.

---

> ### Author Response · Authors · 2023-11-21
> **Response to Reviewer Ywzd**
>
> Thank you for taking the time to review my paper. Following is a point-by-point response:
>
> $\textbf{Reviewer's comment:}$ The results are described in a very confusing manner all over the paper. To quote: "... assume that we have gaps of length $\epsilon$ between consecutive means in our mixture. However, these gaps are isolated, meaning that if $\mu_n+1- \mu_n= \epsilon$ then the adjacent gaps are significantly larger: $\mu_{n+2}-\mu_{n+1}>>1$  and $\mu_{n}-\mu_{n-1}>>1$". What does this mean? On the one hand, you are saying consecutive gaps are small. But immediately after that you are saying certain gaps are significantly large.
>
> $\textbf{Response:}$ The source of confusion here arises from distinguishing between gaps of length $\epsilon$ between consecutive means and consecutive gaps of length $\epsilon$. When I refer to gaps between means, it's important to specify that I am discussing gaps between two consecutive means. For example, in a sequence like $\mu_1< \mu_2< \mu_3< \mu_4<  \mu_5$;  both $\mu_2- \mu_1$   and $\mu_3- \mu_1$   are gaps between means. To clarify, we should first establish that 'gap' refers to the spaces between consecutive means, such as between  $\mu_1$ and $\mu_2$ or $\mu_3$ and $ \mu_4$. Then, we can discuss consecutive gaps.
>
> Consider the example when  $\mu_1=0,  \mu_2= \epsilon, \mu_3=1,  \mu_4=2 $  and $\mu_5= 2+ \epsilon $. In this example, we have two gaps of length $\epsilon$ between $\mu_1$ and $\mu_2$ as well as between $\mu_4$ and $\mu_5$. However, these gaps are not consecutive. If we had $\mu_1=0,  \mu_2=\epsilon, \mu_3=2\epsilon,  \mu_4=1$ and  $ \mu_5= 2$, then we would have two consecutive gaps of length $\epsilon$, one between $\mu_1$ and $\mu_2$ and another between $\mu_2$ and $\mu_3$. A significant portion of the results in my paper is focused on distinguishing between cases like these two. I hope this clarifies things.
>
>
> -$\textbf{Reviewer's comment:}$ Are you considering the univariate or the multivariate Gaussian case? In equation 0.1 you are saying covariance matrix whereas in the our contribution paragraph, you are saying  which corresponds to the univariate case. This is very confusing to read.
>
> $\textbf{Response:}$ Yes, I consider the univariate case, however in the spherical setting  univariate and multivariate are the same. In every dimension you have a mean and the variance is equal to 1. Therefore the $d$-dimensional case transfer to $d$ instances of $1$-dimensional case.
>
> $\textbf{Reviewer's comment:}$ Furthermore, the main contribution seems to be an algorithm for the uniform mixture of spherical Gaussians, which seems rather limited and incremental. Also is it already covered by the prior work? See my comment for Questions.
>
> $\textbf{Response:}$ I will adress this in response to your question.
>
> $\textbf{Reviewer's comment:}$ Now I'll come to the main weakness of the paper. The presentation is extremely poor and unprofessional. For example the paper don't even have a title and section title for the starting section. One cannot simply expect reviewers to spend time on a paper where the authors themselves have not spend much time.
>
> $\textbf{Response:}$ I apologize for problems regarding the writting and the lack of title. I explained the circumstance in my general response to reviewers.
>
> $\textbf{Reviewer's comment:}$ Questions: You are saying the main contribution is an algorithm for the uniform spherical case. I though Hard and Price already show that the tight sample complexity is $\Sigma^{-12}$ which in this case would be, Considering the above fact, why is your result interesting? What am I missing here?
>
> $\textbf{Response:}$ The point you may have missed is that Hardt and Price's results only cover the case when $k=2$, essentially dealing with $2$-mixtures. My work extends their findings to any value of $k$. For instance, if you have data on human height and are informed that there are only two genders, you can use their method to find the mean height for each gender. However, if you are additionally informed that, apart from the two genders, there are also $5$ different races, their method no longer applies because it is designed for $2$-mixtures only. In the latter scenario, you would need to generalize their results to $k=10.$
>
> Furthermore, a significant aspect of their work lies in establishing a relationship between the distances among parameters of the mixture (means and variances) and sample complexity. In their work, with $k=2$, these differences are represented as $\mu_1 - \mu_2$ and $\sigma_1 - \sigma_2$. When attempting to extend their results to accommodate a larger value of $k$, it becomes non-trivial how to generalize this particular aspect. I have addressed this challenge by introducing the pair correlation factor in (0.2).

---

### Official Review · Reviewer_DCj7 · 2023-11-01

**Soundness:** 3 good
**Presentation:** 3 good
**Contribution:** 2 fair
**Rating:** 5
**Confidence:** 2

**Summary:**

This paper studies the sample complexity of learning the parameters of mixtures of (spherical) Gaussians (of equal variance), showing that the distribution of spacing between the means influences the sample complexity.

The proof uses the standard method of moments: first estimate the empirical moments, then compute (an estimate of) the coefficients of the parameter polynomials (whose roots are the means), and finally compute (an estimate of) the means.



Some minor typos:

Page 1
- Title is missing, currently showing Formatting Instructions for ICLR 2024 Conference Submissions
- The paper should point out that in Equation (0.1), weights $\omega_i$ are be non-negative, in addition to sum to 1.
- If the parameters of the mixtures ~all~ **are** well separated, it can be shown that...

Page 5
- “imperial moments we obtain in equation 1.3” should be empirical moments.

Page 9
- References 6 and 7 are the same.

Throughout
- Left (double) quotes should use `` instead of '' when typesetting in $\LaTeX$, so that we get “this” instead of ”this” (note the left quote).

**Strengths:**

The upper bounds generalize previous results of 2 Gaussians by Hardt and Price to $k$ (spherial) Gaussians (of equal variance), and matches their lower bounds.

Section 0.1 explaining Pair Correlation, with the two examples without and with consecutive gaps, is helpful to understanding to this central concept for the main result. Besides, identifying the role of the Pair Correlation of means is itself a contribution, explaining the dependence of sample complexity to distribution of parameters (means).

**Weaknesses:**

Works only for the mixture of spherical Gaussians of equal variance, and for the mean zero case, depends on a new parameter of pair correlation of means, which limits applications of this result.

**Questions:**

Besides theoretical interests, may the authors suggest applications of the results, considering their limitations (works only for the spherical case of equal variance, and depends on pair correlation of means)?

---

> ### Author Response · Authors · 2023-11-21
> **Response to Reviewer DCj7**
>
> Thank you for taking the time to review my paper. In response to your comments on its weaknesses, please note that when we consider a scenario where all variances are equal, it represents the most challenging case for our problem. Essentially, when the set of means is fixed, if all variances are equal, we encounter the worst sample complexity. Therefore, the sample complexity we derive serves as an upper bound applicable to a broader context where variances are not necessarily equal. It's also important to mention that Hardt and Price [1] have previously demonstrated that for 2-mixtures, when means are in close proximity, differences in variances would reduce the sample complexity.
>
> My results can be particularly valuable to researchers and practitioners dealing with the parameter estimation of GMMs. Especially when there is some prior information available about the distribution of spacings between means. I have elaborated further on this in my responses to your questions and in the reply to reviewer EBv4.
>
> As for assuming the mean of the mixture is zero, we can always center our data by subtracting the empirical mean from each data point. This adjustment allows us to work in a setting where the overall mean is zero.
>
> I would also like to draw your attention to two recent papers [2,3] that delve into the spherical case, assuming an additional condition of means being separated by (log k). These papers explore even more specific instances of the problem I studied and have been published in respected ML journals. This underscores that this is an active area of research in the theoretical ML. Nevertheless, I acknowledge that one limitation of my work is the assumption of equal variances.
>
> In response to your question regarding the practical applications, I have addressed a portion of this in my response to referee EBv4. As previously mentioned, when we fix the set of means, the most challenging sample complexity arises when all variances are equal. Therefore, my findings indicate that the sample complexity in a general setting (where variances are not equal, necessarily) is lower than that in the spherical case. In real-world applications, assume we have some knowledge about the distribution of means, even if we do not know their specific locations. Assume we have reason to believe they follow a distribution such as the Spatial Poisson point process. This assumption suggests the existence of small gaps between means. Using alternative methods, a factor of $(1/\text{minimum gap})^k$ contributes to the sample complexity. In contrast, my method introduces a factor of $(1/\text{minimum gap})^E$, where E represents the expectation of consecutive gaps with minimum distance. In practice, E tends to be much smaller than k. Therefore, at minimum, using my approach we know how many samples carry enough information to detect parameters of the mixture, even if the algorithm does not work for the general case.
>
> References:
>
> [1] Moritz Hardt, Eric Price, STOC '15: Proceedings of the forty-seventh annual ACM symposium on Theory of Computing. https://doi.org/10.48550/arXiv.1404.4997
>
> [2] Mingda Qiao, Guru Guruganesh, Ankit Singh Rawat, Avinava Dubey, Manzil Zaheer. "A Fourier Approach to Mixture Learning." To appear at NeurIPS 2022.
>
> [3] Allen Liu and Jerry Li. 2022. "Clustering mixtures with almost optimal separation in polynomial time." In Proceedings of the 54th Annual ACM SIGACT Symposium on Theory of Computing (STOC 2022).

---

> > ### Comment · Reviewer_DCj7 · 2023-11-22
> >
> > I acknowledge and appreciate the response by the authors. However, I keep my scores given the current state of the paper.

---

### Author Response · Authors · 2023-11-21
**General comment to reviewers**

First, I would like to apologize to all referees for the shortcomings in the submission, including the absence of a title and issues with the writing. I was unaware of the strict page limit imposed by ICLR, and Open Review sent a reminder email just 12 hours before the submission deadline. Unfortunately, due to the time zone I am in, I received this email with only 5 hours remaining until the deadline.

In a rush to meet the requirement, I began to trim paragraphs from the paper. Regrettably, there was insufficient time to fully maintain the coherency of the paper. I have reviewed papers, and I understand the frustration when submissions are not presented properly. Typically, I do my best to write properly, but this time, I was caught off-guard by both the page limit and the time constraint. I understand that it is my responsibility to review the requirements of each conference. This is just to explain that the issues with the paper's writing were not indicative of my disregard for the valuable time of the reviewers.

I also want to express my gratitude for taking the time to review our paper. Your feedback has been very useful, and I genuinely appreciate the insights and suggestions you have provided

---

### Meta-Review · Area_Chair_FUXp · 2023-12-08

**Metareview:**

Gaussian mixtures are extensively well-studied, with hundreds of results in the literature. Due to a lack of discussion and comparison with this related work, the initial reviews for this paper were understandably negative. During the discussion, the authors helped to clarify their contributions, but it was still clear that a significant (possibly complete) rewrite of the paper is needed to address related work in detail. For this reason, the paper is not ready to be accepted in its current form.

**Justification For Why Not Higher Score:**

See meta-review

**Justification For Why Not Lower Score:**

N/A

---

### Decision · Program_Chairs · 2024-01-16

Reject